# Cryopreservation method for *Drosophila melanogaster* embryos

Li Zhan[1,2], Min-gang Li[3], Thomas Hays [3✉] & John Bischof[1,2,4✉]

The development of a widely adopted cryopreservation method remains a major challenge in *Drosophila* research. Here we report a robust and easily implemented cryopreservation protocol of *Drosophila melanogaster* embryos. We present innovations for embryo permeabilization, cryoprotectant agent loading, and rewarming. We show that the protocol is broadly applicable, successfully implemented in 25 distinct strains from different sources. We demonstrate that for most strains, >50% embryos hatch and >25% of the resulting larvae develop into adults after cryopreservation. We determine that survival can be significantly improved by outcrossing to mitigate the effect of genetic background for strains with low survival after cryopreservation. We show that flies retain normal sex ratio, fertility, and original mutation after successive cryopreservation of 5 generations and 6-month storage in liquid nitrogen. Lastly, we find that non-specialists are able to use this protocol to obtain consistent results, demonstrating potential for wide adoption.

---

[1] Department of Mechanical Engineering, University of Minnesota, Minneapolis, MN, USA. [2] Center for Advanced Technologies for the Preservation of Biological Systems (ATP-Bio), University of Minnesota, Minneapolis, MN, USA. [3] Department of Genetics, Cell Biology and Development, University of Minnesota, Minneapolis, MN, USA. [4] Department of Biomedical Engineering, University of Minnesota, Minneapolis, MN, USA. ✉email: haysx001@umn.edu; bischof@umn.edu

The fruit fly (*Drosophila melanogaster*), a foundational genetic model organism for biological research in the past century, has driven important discoveries leading to countless biomedical science breakthroughs including six Nobel Prizes[1–6]. Not surprisingly, there are >160,000 unique genotypes held in individual research laboratories and stock centers worldwide and this number is growing[7]. Currently, the stocks must be manually maintained through frequent and costly transfer of breeding adults to fresh food. In comparison, cryopreservation of *Drosophila melanogaster* provides enormous advantages, including protection against genetic drift to stabilize genotypes and allow expanded genetic and evolutionary studies, decreasing stock maintenance costs, and reducing the risk of stock loss caused by contamination or accidental mixing of precious stocks[7–9]. Indeed, other investigators have sought to develop methods to cryopreserve *Drosophila* embryos using only wildtype strain (i.e., *Oregon R*)[8,9]. However, these protocols failed to be adopted due to the lack of reproducibility and protocol complexity requiring specialized devices (Supplementary Table 1, Supplementary Fig. 1); Moreover, no protocol has been successfully implemented to cryopreserve a broad collection of *Drosophila* stocks, both wildtype and mutant[7]. Thus, after more than a century of *Drosophila* research there is no simple, robust, and universal cryopreservation method available to the *Drosophila* community[7].

The major challenges to cryopreserve *Drosophila melanogaster* embryos include cryoprotectant agent (CPA) loading, vitrification with scalability, embryo age-dependent survival, and strain-dependent genetic backgrounds. The first hurdle is to introduce CPA directly into the embryo. After dechorionation, the embryo is impermeable to CPA due to the waxy layer and vitelline membrane[10,11]. Assuming CPA can be loaded, rapid cooling and warming rates are required to achieve cryopreservation via vitrification, a solidification process from liquid into glass with minimal lethal ice formation[12,13]. However, it is challenging to scale up the conventional vitrification tools to handle large numbers of *Drosophila* embryos (i.e., >1000) while maintaining rapid cooling and warming rates (Supplementary Table 2). In addition, while embryo age was reported to significantly affect cryopreservation outcomes, little guidance was provided to identify and reproducibly obtain the optimal age for non-specialists[8,9,14,15]. Finally, genetic variation in other cell types has been shown to lead to variable tolerance to CPA toxicity[16–18]. This suggests that the diverse genetic background of *Drosophila* may impact cryopreservation success although this is largely unstudied.

Here, to tackle the above mentioned challenges, we introduce innovations especially focusing on permeabilization, CPA loading and vitrification, leading to a simple and robust cryopreservation protocol supporting wide adoption by the *Drosophila* community (Supplementary Table 1).

## Results

**Embryo stage optimization**. We first performed extensive optimization on each step of the protocol using a stock from our lab named M2 (Fig. 1a). As a derivative of *w[1118]*, M2 carries a traceable single nucleotide polymorphism (SNP) on the X chromosome and is homozygous, viable, and fertile. As the embryonic development rate is highly temperature dependent[19], we established a robust procedure to stage the embryos by combining the chronological age via strict control of incubation time at a set incubation temperature (i.e., 20.1 ± 0.05 °C, Supplementary Fig. 2), and the morphological features via inspection of embryo gut appearance under the compound and/or dissecting microscopes (Fig. 1b). Specifically, under the compound microscope, the gut appeared as dark structures (white outlines were manually added to the images for enhanced clarity, Fig. 1b). Under the dissecting microscope, the gut appeared a milky color (Fig. 1b lower panels). From 19 h to 24 h, the appearance of the gut changes from a heart-like shaped structure (19 h) to a set of 3–4 semi-parallel bars that lie orthogonal to the embryo long axis (20 h), that becomes progressively more tilted (21–22 h) and eventually morphs into a more extended shape (23–24 h). After cryopreserving embryos collected at various time points, or age, we evaluated embryo survival by the hatch rate (embryos to larvae) and adult survival rate (hatched larvae, pupate, then eclose to adults). We established that 22 h old embryos provided the highest post cryopreservation survival, which corresponds to early stage 16 when head involution and dorsal closure have been completed (Fig. 2a)[20]. For embryos at older ages, the impermeable cuticle layer starts to form, precluding the uptake of CPA and therefore survival decreased sharply[9]. The age of flies used for embryo collection also impacted the cryopreservation outcome. A lower adult survival rate was observed for embryos collected using older flies (9–12 days) than young ones (1–4 days), potentially due to female egg retention. Such retention broadens the age distribution of embryos in a collection and lowers the number of embryos at the proper developmental age for cryopreservation (Supplementary Fig. 3).

**Embryo permeabilization**. As a critical step, we then employed a simple mesh basket to perform permeabilization using the mixture of D-limonene and heptane (LH) (Supplementary Fig. 1, details in the Supplementary Materials). In contrast, previous publications used a complex protocol and specialized device that are difficult to implement and leads to an inconsistent outcome. We found that 10 s soaking time in the LH solution was adequate for wax removal and permeabilization of the vitelline membrane, while causing minimal injury (Supplementary Fig. 4). Permeabilized embryos stained red when soaked in rhodamine B solution and were stripped of the wax layer as visualized by electron microscopy (Fig. 1c, Supplementary Fig. 5). In general, the LH-treated embryos are permeable to CPAs including ethylene glycol (EG), propylene glycol (PG), and dimethyl sulfoxide (DMSO), but are not permeable to CPAs including the sugars such as sucrose, sortibol, and trehalose[17,21]. To introduce CPA into the embryos for subsequent vitrification, a monolayer of embryos was initially exposed to low concentration permeable CPA (i.e., 13 wt %). More than 90% of the embryos first lost water and shrank due to higher external osmolarity, followed by swelling as CPA slowly entered until reaching equilibrium (Fig. 1c).

**CPA loading and optimization**. At this point, intra-embryonic CPA concentration was elevated through dehydration by placing the embryos in a high concentration CPA (i.e., ~39 wt%) at 4 °C. Dehydrated embryos appeared flat in shape with multiple "wrinkles" on the surface (Fig. 1c). The final intra-embryonic CPA concentration is a function of dehydration time, total osmolarity, and permeable CPA concentration used for this dehydration step. Higher intra-embryonic CPA concentration results in greater protection against lethal ice formation during ensuing cooling and rewarming, but also can lead to greater CPA toxicity especially at suprazero temperatures. To achieve the optimal balance, we compared the CPA toxicity (i.e., post dehydration survival) and post cryopreservation survival using different dehydration times, dehydration CPA concentrations and dehydration CPA compositions. Under the same weight concentration, EG has proven to have the least CPA toxicity and highest survival post cryopreservation (Supplementary Fig. 6). On the other hand, DMSO has the highest CPA toxicity shown in Supplementary Fig. 6 perhaps due to the neurotoxicity[22,23]. In

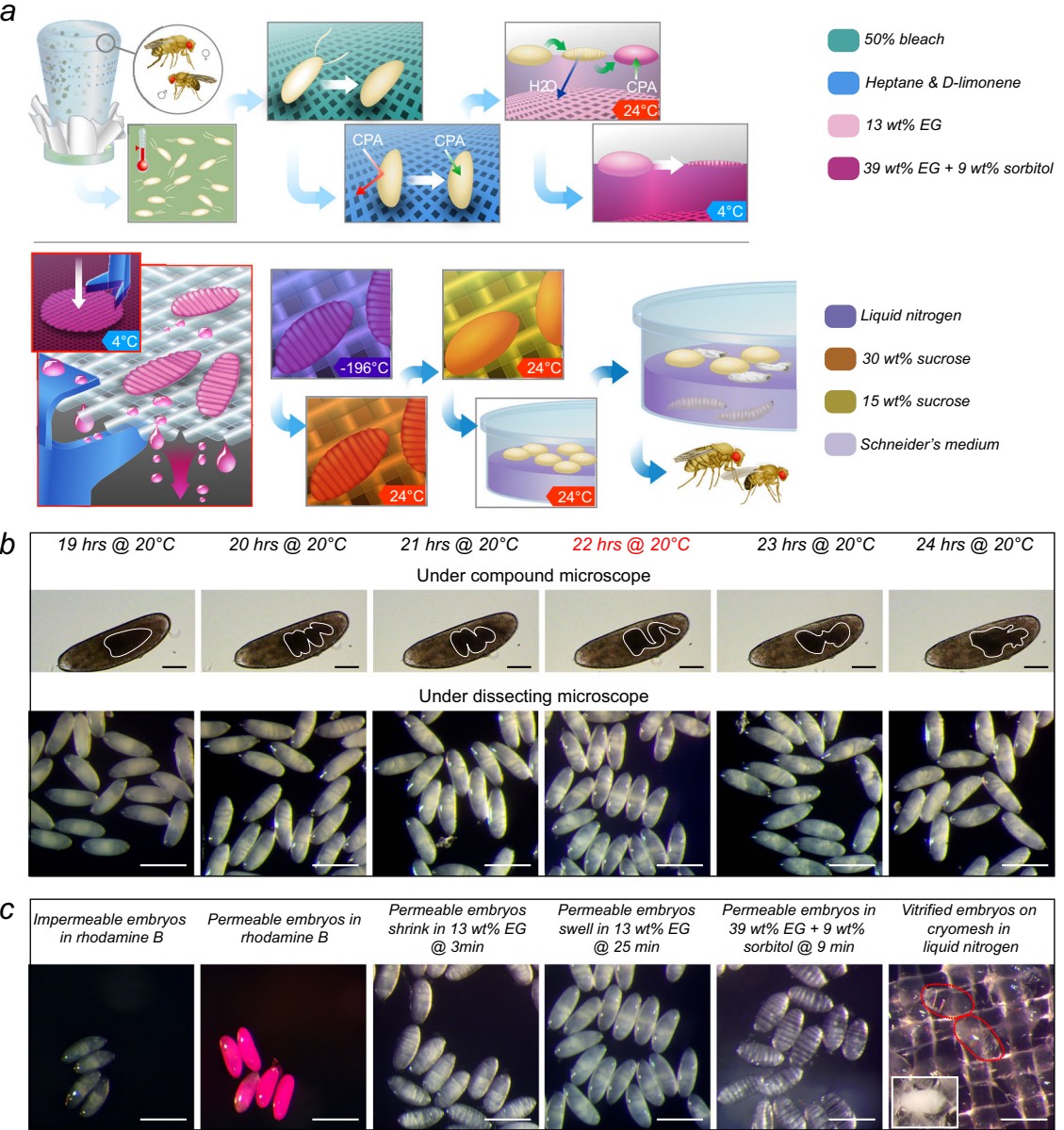

**Fig. 1 Schematic overview of cryopreservation procedures for *Drosophila melanogaster* embryos and detailed pictorial illustration for critical steps.**
**a** On day 1, embryos were collected on a grape juice plate for 1 h period at room temperature (24 °C), then placed in a 20 °C incubator until reaching the desired stage for cryopreservation. On day 2, embryos were first dechorionated and permeabilized, followed by CPA loading and dehydration. The cryomesh was used to pick up the dehydrated embryos for vitrification, storage, rewarming, and CPA unloading. Afterward, embryos were cultured in Schneider's medium overnight. On day 3, hatched larvae were transferred to food vials until adult emergence. **b** Images of embryo gut morphology under dissecting and compound microscopes after different incubation time at 20 °C. **c** Images of embryos at different steps during cryopreservation. From left to right, impermeable and permeable embryos after 5 min in 0.1% rhodamine B; permeable embryos first shrunk then swelled in 13 wt% EG; flattened embryos after dehydration; vitrified embryos appeared transparent in LN$_2$ (two of them were outlined in red), inset is a crystallized embryo (i.e., failure). Five experiments were repeated independently with similar results for (**b–c**). Scale bars in compound microscope images are 100 μm, in dissecting microscope images are 500 μm.

addition, the use of permeable CPA cocktails to infiltrate permeabilized embryos for vitrification did not outperform the individual CPAs (Supplementary Fig. 6). However, a combination of permeable and non-permeable CPAs does reduce CPA toxicity and provides superior post cryopreservation survival, compared to permeable CPAs alone with the same total osmolarity (Fig. 2b, Supplementary Fig. 7). Further, when 39 wt% EG + 9 wt% sorbitol was used as the dehydration CPA, post cryopreservation survival remained similar with increasing dehydration time from 9 min to 21 min (Supplementary Fig. 8). Replacing sorbitol with sucrose or trehalose did not affect post cryopreservation survival

(Supplementary Fig. 9). Altogether, 9 min dehydration in 39 wt% EG + 9 wt% sorbitol was selected to reduce the cost of the reagents and minimize the time of the protocol.

**Vitrification and rewarming**. To cryopreserve embryos in large quantities, we developed a cryomesh approach–a nylon mesh attached to a thin polystyrene holder. A 2 cm by 2 cm size mesh can easily accommodate ~1700 embryos (details in Supplementary Materials). We used 200–600 embryos for each experiment in this study. Almost all of the floating embryos on the

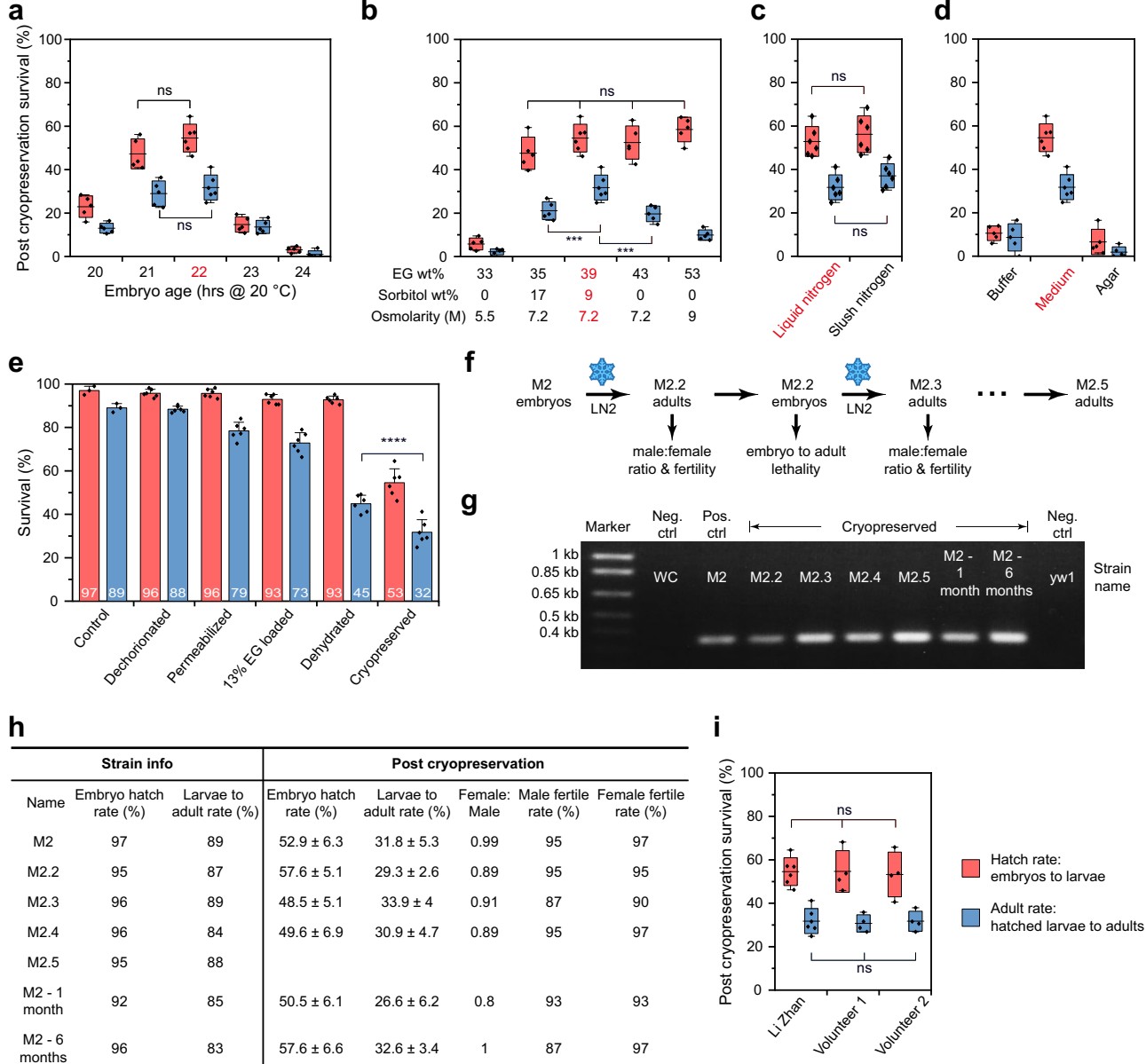

dehydration CPA solution are transferred to the cryomesh within seconds by pressing a dry cryomesh into the CPA solution and lifting it out (Fig. 1a). Importantly, wicking the remaining CPA solution off the cryomesh prior to vitrification reduced the mass on the cryomesh by 10-fold, significantly improving the cooling and warming rates, as well as post cryopreservation survival (Fig. 3a-d). This "CPA solution free" method maximizes the cooling and warming rate while allowing the processing of large numbers of embryos thereby outperforms traditional vitrification tools (Fig. 3c, Supplementary Table 2). The cryomesh with the embryos was then quickly plunged into liquid nitrogen (LN$_2$) for vitrification and was stored in LN$_2$ until future use. Vitrified embryos appeared transparent in LN$_2$ while crystallized embryos (i.e., failure) looked white (Fig. 1c, Supplementary Fig. 10). Slush nitrogen (SN$_2$) made by a specialized device was required in previous publications and also tested here[8,9,24]. A thermocouple was placed in contact with the embryos on a cryomesh and recorded a faster cooling rate in SN$_2$ than LN$_2$ but a similar warming rate compared with LN$_2$ (Supplementary Fig. 11). Further, we recorded similar post cryopreservation survival rates

using LN$_2$ or SN$_2$, which allows us to substantially simplify the cryopreservation procedure by using LN$_2$ (Fig. 2c).

The removal of CPA solution on the cryomesh provides a high warming rate consistently, therefore improving the robustness of our protocol. Numerous studies suggest that a high warming rate is the vital step in vitrification-based cryopreservation and can even "rescue" poorly cooled biomaterials with a certain amount of ice present[25–27]. Heat transfer modeling indicates a dramatically lower warming rate when embryos were surrounded by CPA solution (Supplementary Fig. 12). For instance, the warming rate drops to $2.4 \times 10^4$ °C/min with 250 μm thickness CPA layer (Fig. 2e-f). Similar warming rates were reported in the previous protocols (Supplementary Table 2)[8]. After removing the CPA solution (Fig. 3a), our modeling suggests that the larger the contact area between the embryo and cryomesh, the faster the embryos rewarm as a consequence of the nylon mesh rewarming faster than the embryos (Fig. 3g-h, Supplementary Fig. 13). Modeling implies the average warming rates of embryos with minimum and maximum mesh contact was $2.2 \times 10^5$ °C/min, consistent with the experimental measurement (Fig. 3c). In

**Fig. 2 Cryopreservation protocol optimization and post cryopreservation evaluation using a strain named M2.** Optimization of (**a**) embryos age, (**b**) dehydration CPA compositions, (**c**) different phase of nitrogen (liquid vs. slush), and (**d**) post cryopreservation embryo culture methods. Post cryopreservation survival is evaluated by the hatch rate (embryos to larvae) and adult rate (larvae, pupate, then eclose to adults). For (**a–d**), the optimal conditions are labeled in red and a shared y axis is used. **e** Survival after each step of the cryopreservation process. **f** Flow chart for evaluation of male to female ratio, fertility, and lethality post cryopreservation across multiple generations. **g** PCR confirmed the original mutation in M2 was maintained after cryopreservation of multiple generations and different storage time in $LN_2$. Three experiments were repeated independently with similar results. **h** Post cryopreservation evaluation after multiple generations and different storage time in $LN_2$. **i** Two volunteers were trained to perform the cryopreservation. The red/blue hatch rate/adult rate labeling applies to the entire Fig. 2. Box and horizontal line represent standard deviation and mean respectively, whiskers represent max and min. Error bars represent standard deviation in (**e**). $n = 5$ or 6 for (**a–d**); $n = 3$ or 6 for (**e**); $n = 4$ or 6 for (**i**); $n$ stands for independent replicates. In (**a**), for the embryo age of 20 h, 1444 embryos were pooled over $n = 5$ independent replicates; for the embryo age of 21 h, 1431 embryos were pooled over $n = 5$ independent replicates; for the embryo age of 22 h, 1695 embryos were pooled over $n = 6$ independent replicates; for the embryo age of 23 h, 1567 embryos were pooled over $n = 5$ independent replicates; for the embryo age of 24 h, 1340 embryos were pooled over $n = 5$ independent replicates. In (**b**), for 33 wt% EG, 1587 embryos were pooled over $n = 5$ independent replicates; for 35 wt% EG + 17 wt% sorbitol, 1374 embryos were pooled over $n = 5$ independent replicates; for 39 wt% EG + 9 wt% sorbitol, 1695 embryos were pooled over $n = 6$ independent replicates; for 43 wt% EG, 1531 embryos were pooled over $n = 5$ independent replicates; for 53 wt% EG, 1466 embryos were pooled over $n = 5$ independent replicates. In (**c**), for liquid nitrogen, 1695 embryos were pooled over $n = 6$ independent replicates; for slush nitrogen, 1825 embryos were pooled over $n = 6$ independent replicates. In (**d**), for buffer, 1368 embryos were pooled over $n = 5$ independent replicates; for medium, 1695 embryos were pooled over $n = 6$ independent replicates; for agar, 1446 embryos were pooled over $n = 5$ independent replicates. In (**e**), for control, 1245 embryos were pooled over $n = 3$ independent replicates; for dechorionated, 1963 embryos were pooled over $n = 6$ independent replicates; for permeabilized, 1551 embryos were pooled over $n = 6$ independent replicates; for 13% EG loaded, 1865 embryos were pooled over $n = 6$ independent replicates; for dehydrated, 1984 embryos were pooled over $n = 6$ independent replicates; for cryopreserved, 1695 embryos were pooled over $n = 6$ independent replicates; In (**i**), for Li Zhan, 1695 embryos were pooled over $n = 6$ independent replicates; for volunteer 1, 1315 embryos were pooled over $n = 4$ independent replicates; for volunteer 2, 1256 embryos were pooled over $n = 4$ independent replicates. Two-sided multivariate analysis of variance (MANOVA) and Tukey's post hoc were used for statistical analysis. ns, $p > 0.05$; ***$p \leq 0.001$; ****$p \leq 0.0001$.

addition, the modeling reveals similar warming rates throughout each embryo (Fig. 3h).

**CPA removal**. For intra-embryonic CPA removal after rewarming, dehydrated embryos were exposed to 15 wt% sucrose solution prior to the cryobuffer (i.e., a isotonic saline buffer) to mitigate the osmotic shock. We also tested direct unloading in the cryobuffer, which surprisingly showed a similar embryo hatch rate, but slightly lower adult survival rate (Supplementary Fig. 14). This likely indicates that the vitelline membrane helps to avoid overswelling of the dehydrated embryos (Supplementary Fig. 9b). Further, we demonstrated that the cost of cryopreservation can be greatly reduced by using a defined cryobuffer as the carrier solution to prepare CPA and unloading solutions. Use of a more expensive Schneider culture medium as the carrier solution did not elevate post cryopreservation survival (Supplementary Fig. 15). We then tested different embryo culture methods post cryopreservation, since embryos are vulnerable to an external environment post permeabilization (Fig. 2d). For this step, floating embryos on Schneider medium provided the best survival rate by comparison to floating embryos on the cryobuffer, or placing the embryos on agar plates as previously reported[9]. Indeed, Schneider medium supplies essential nutrients for further development and an aqueous environment for continuous unloading of intra-embryonic CPA. Using the optimal cryopreservation protocol, stepwise survival of strain M2 is presented in Fig. 3e. After cryopreservation, the hatch rate and adult rate were $52.9 \pm 6.3$ % and $31.8 \pm 5.3$ %, compared to 97% and 89% for untreated embryos.

**Protocol robustness evaluation**. Further, we assessed the ease of application and robustness of the protocol. First, we trained two non-specialist volunteers, including one high school student. Both volunteers obtained consistent post cryopreservation survival rates (Fig. 2i). In addition, we completed a more extensive analysis of the cryopreservation of M2, addressing the impacts of $LN_2$ storage time and repetitive cycles of cryopreservation. We observe similar post cryopreservation survival rates for M2 embryos

regardless of storage in $LN_2$ for 1 min, 1 month, 6 months, or 1 year (Supplementary Fig. 16). To investigate the impact of repeated cryopreservation cycles, we recovered adults (M2.2) from the first cryopreserved embryos and subsequently repeated the collection and cryopreservation of embryos for multiple generations (i.e., M2.2–M2.5, Fig. 2f). PCR analysis demonstrated that the SNP present in the original parental M2 strain was maintained in all the subsequent progenies recovered following cryopreservation (Fig. 2g). In addition, all the progenies showed similar embryo to adult survival rates compared to the original M2 strain with no generational decline in survival rates post cryopreservation (Fig. 2h). An equal sex ratio among the progeny suggests that no lethal mutations were introduced on the X chromosome after repeated cryopreservations or long-term $LN_2$ storage (Fig. 2h). In overview, comparable post cryopreservation survival and fertility were retained across multiple generations, for different $LN_2$ storage times, and in the hands of non-specialists, demonstrating the simplicity and translatability of our protocol.

**Application to 24 distinct strains**. Finally, we validated our protocol with 24 other strains. These included wildtype, mutant, single balancer and double balancers from different sources including the Bloomington Stock Center, our lab and other *Drosophila* labs (Fig. 4). To investigate whether the optimized conditions for M2 applies to other strains, we tested each variable with at least two other strains (Supplementary Table 3). We show that the same optimal conditions apply across strains (Supplementary Figs. 17-25), although some variation was observed for strains exhibiting variable rates of embryo development. Specifically, for strain S7, 21 h old embryos provided higher post cryopreservation survival than 22 h old embryos due to slightly faster embryonic developmental rate or increased egg retention time (Supplementary Fig. 26). In addition, as genetic crosses are routinely performed in *Drosophila* labs, we derived new strains by crossing them to explore the impact on cryopreservation outcome. For example, WC1b was generated by crossing a single *w [1118]* male to S2 strain to isogenize the X chromosome. Figure. 4 showed the summary of the post cryopreservation survival

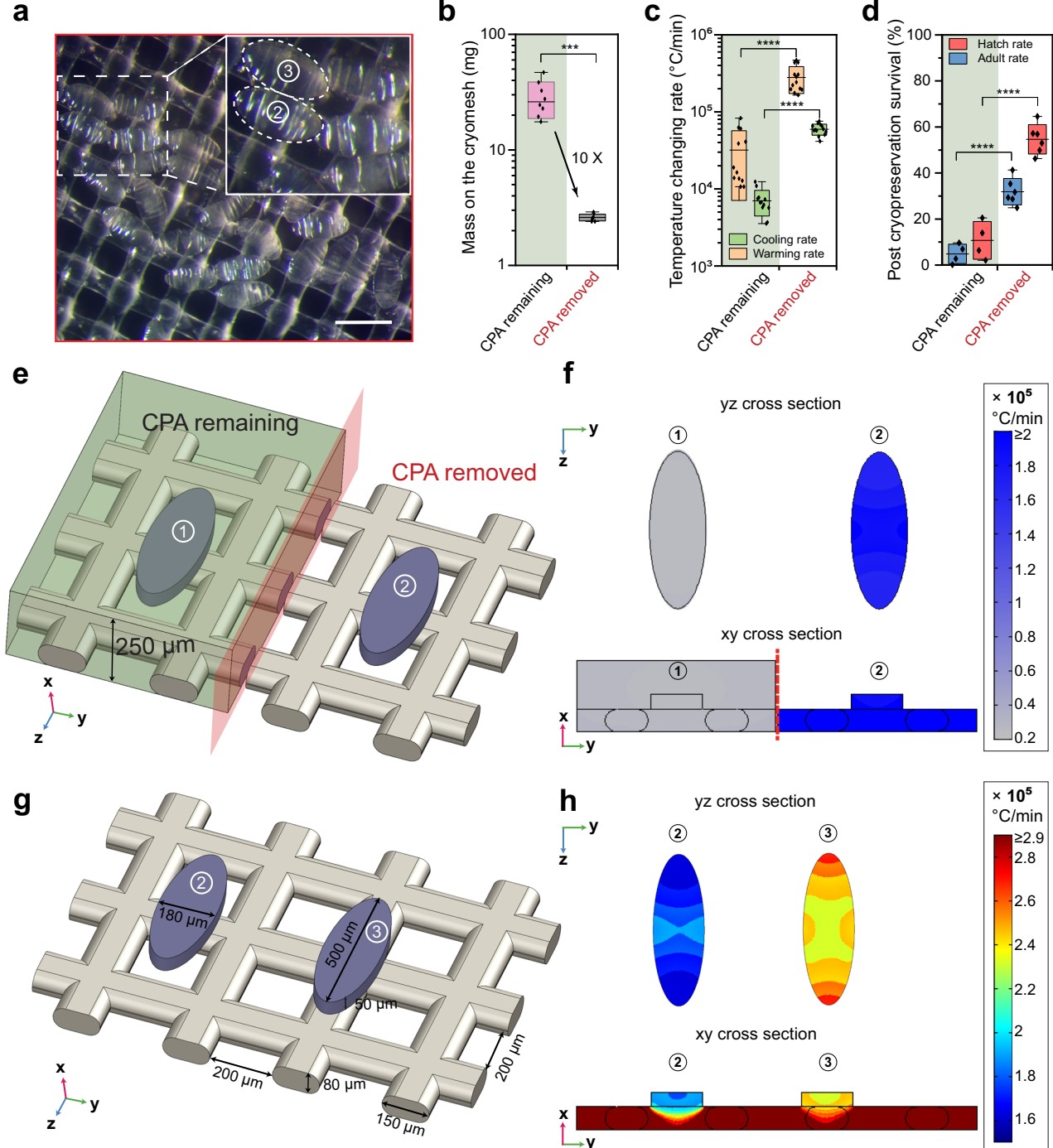

**Fig. 3 Cooling and warming rate using the cryomesh. a** Image of embryos on the cryomesh with CPA removed. Inset is the embryo with minimal (embryo 2) or maximal (embryo 3) contact with the cryomesh (outlined in white). Five experiments were repeated independently with similar results. **b** After the embryos were collected, the weight on the cryomesh was measured with CPA removed or remaining. **c** Measured cooling and warming rate of the embryos on the cryomesh. **d** Post cryopreservation survival with CPA removed or remaining on the cryomesh. **e** Warming rate modeling for the comparison of CPA remaining (embryo 1) or removed (embryo 2) on the cryomesh. **f** Simulated warming rates at different cross-sections through the center point of embryo 1 and embryo 2. **g** With CPA removed, warming rate modeling for embryo with minimal (embryo 2) or maximal (embryo 3) contact with the cryomesh. **h** Warming rates at different cross-sections through the center point of embryo 2 and embryo 3. In (**f**) & (**h**), color scales represent the simulated warming rates. Scale bar is 500 μm. Box and horizontal line represent standard deviation and mean respectively; whiskers represent max and min. $n = 8$ for (**b**); $n = 12$ for (**c**); $n = 4$ or 6 for (**d**); $n$ stands for independent replicates. In (**d**), 1695 embryos were pooled over $n = 6$ independent replicates; 1433 embryos were pooled over $n = 4$ independent replicates. Two-sided multivariate analysis of variance (MANOVA) and Tukey's post hoc were used for statistical analysis. ***$p \leq 0.001$; ****$p \leq 0.0001$.

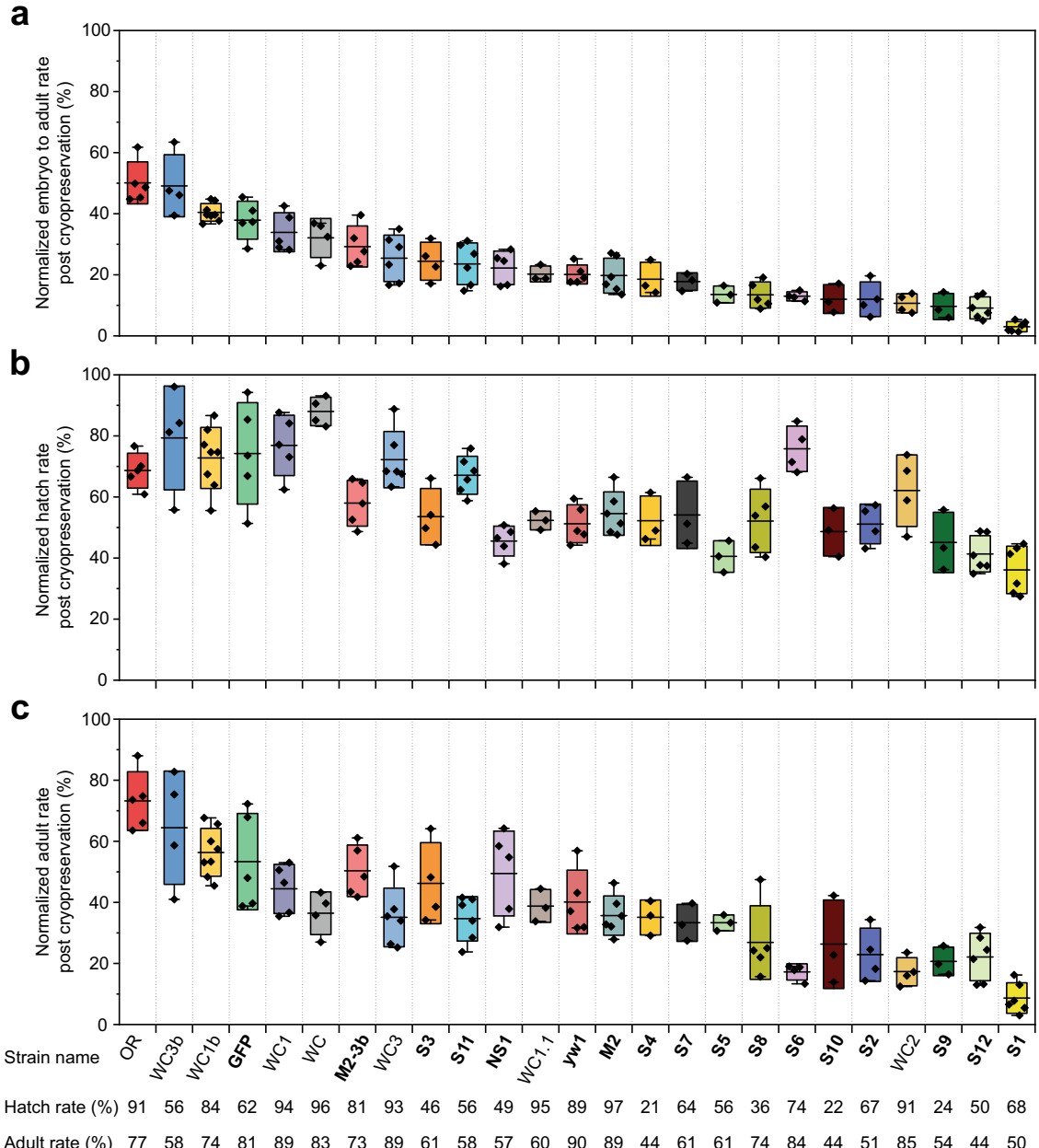

| Strain name | OR | WC3b | WC1b | GFP | WC1 | WC | M2-3b | WC3 | S3 | S11 | NS1 | WC1.1 | yw1 | M2 | S4 | S7 | S5 | S8 | S6 | S10 | S2 | WC2 | S9 | S12 | S1 |
|---|---|---|---|---|---|---|---|---|---|---|---|---|---|---|---|---|---|---|---|---|---|---|---|---|---|
| Hatch rate (%) | 91 | 56 | 84 | 62 | 94 | 96 | 81 | 93 | 46 | 56 | 49 | 95 | 89 | 97 | 21 | 64 | 56 | 36 | 74 | 22 | 67 | 91 | 24 | 50 | 68 |
| Adult rate (%) | 77 | 58 | 74 | 81 | 89 | 83 | 73 | 89 | 61 | 58 | 57 | 60 | 90 | 89 | 44 | 61 | 61 | 74 | 84 | 44 | 51 | 85 | 54 | 44 | 50 |

**Fig. 4 Normalized post cryopreservation survival of 25 strains using the universal cryopreservation protocol. a** Normalized embryo to adult rate of the 25 strains with different genotypes after cryopreservation. **b** Normalized hatch rate (embryos to larvae) and **(c)** adult rate (larvae, pupate, then enclose to adults) of the 25 strains after cryopreservation. The strain name is listed in (**c**). The hatch rate and adult rate of the control embryos (without treatment) are listed and used to normalize the post cryopreservation survival. In total, 17 mutant strains were included (strain name bolded). Strain GFP was obtained from the Bloomington Stock Center (stock # 30877). Strain S11 was obtained from the other *Drosophila* lab. $n = 3$ to 8 independent replicates for various strains. For strain OR, 1881 embryos were pooled over $n = 5$ independent replicates; for strain WC3b, 1267 embryos were pooled over $n = 4$ independent replicates; for strain WC1b, 2519 embryos were pooled over $n = 8$ independent replicates; for strain GFP, 1859 embryos were pooled over $n = 5$ independent replicates; for strain WC1, 1439 embryos were pooled over $n = 5$ independent replicates; for strain WC, 1874 embryos were pooled over $n = 4$ independent replicates; for strain M2-3b, 1612 embryos were pooled over $n = 5$ independent replicates; for strain WC3, 1741 embryos were pooled over $n = 6$ independent replicates; for strain S3, 1006 embryos were pooled over $n = 4$ independent replicates; for strain S11, 1500 embryos were pooled over $n = 6$ independent replicates; for strain NS1, 1478 embryos were pooled over $n = 5$ independent replicates; for strain WC1.1, 1183 embryos were pooled over $n = 3$ independent replicates; for strain yw1, 1971 embryos were pooled over $n = 5$ independent replicates; for strain M2, 1695 embryos were pooled over $n = 6$ independent replicates; for strain S4, 991 embryos were pooled over $n = 3$ independent replicates; for strain S7, 1077 embryos were pooled over $n = 3$ independent replicates; for strain S5, 1063 embryos were pooled over $n = 3$ independent replicates; for strain S8, 1564 embryos were pooled over $n = 5$ independent replicates; for strain S6, 1112 embryos were pooled over $n = 4$ independent replicates; for strain S10, 920 embryos were pooled over $n = 3$ independent replicates; for strain S2, 1119 embryos were pooled over $n = 4$ independent replicates; for strain WC2, 1664 embryos were pooled over $n = 4$ independent replicates; for strain S9, 960 embryos were pooled over $n = 3$ independent replicates; for strain S12, 1644 embryos were pooled over $n = 6$ independent replicates; for strain S1, 1817 embryos were pooled over $n = 6$ independent replicates. Box and horizontal line represent standard deviation and mean respectively; whiskers represent max and min.

normalized by embryos without any treatment. Although strain-dependent survival was noted, a normalized embryo to adult survival rate higher than 10% can be achieved in the majority of strains (Fig. 4, Supplementary Table 4). A second chromosome balancer stock S1 yielded very low embryo to adult survival rates. To investigate whether the genetic background variations of S1 caused this low survival rate, we outcrossed S1 to the GFP strain that exhibits a higher survival rate post cryopreservation. The resultant strain, NS1, retained its second chromosome balancer, yet showed improved post cryopreservation survival (Fig. 4, Supplementary Table 4), demonstrating that survival rates can be improved by outcrossing to mitigate genetic background factors that impact cryopreservation.

## Discussion

To explore factors underlying the strain-dependent survival following cryopreservation, we first examined the contribution of embryo age distribution. One hour embryo collections from different strains were incubated at 24 °C and the hatch frequency at various times was recorded (Supplementary Fig. 26). We observed that strains M2, WC, and GFP showed a narrow embryo age distribution in timed embryo collections while strains S1, NS1, and S7 have a broader distribution. In fact, various egg retention patterns regulated by genetics have been reported[28,29]. As post cryopreservation survival depends on embryo age upon vitrification, strains with limited egg retention (i.e., narrow embryo age distribution) could potentially have higher post cryopreservation survival rates. Besides genetic variation, we showed that a clutch of embryos from older parent flies display a broader range of embryo ages in timed collections, than did embryos collected from younger parent flies (Supplementary Fig. 3). In the case of S1 and NS1, post cryopreservation survival still varied despite a similar broad embryo age distribution (Supplementary Fig. 26). Analysis of the stepwise survival during the cryopreservation procedure indicates that the genetic variation between S1 and NS1 results in discrepant tolerance to CPA toxicity (Supplementary Fig. 27). The potential genetic factors behind those results remain unknown and will require further study.

In summary, we have developed a simple and robust cryopreservation method for *Drosophila* embryos that shows good survival rates and great promise for wide adoption by individual *Drosophila* labs and stock centers. After the embryos reach an optimal stage, our cryopreservation protocol can process large numbers (>1000) of embryos within 1 h (i.e., ~35 min for vitrification and storage in $LN_2$, ~25 min for rewarming and CPA unloading) using one cryomesh. To adopt our protocol for any new lab strain, we suggest the flowchart shown in Supplementary Fig. 28, using one of the high survival strains reported here as a positive control. For strains whose initial cryopreservation survival rates are unsatisfactory (i.e., strain S1), we also present a solution by outcrossing such strains to stocks that show high post cryopreservation survival. Further investigation focusing on the genetic aspect could potentially identify the gene(s) that contributes to the variation in post thaw survival, therefore allowing targeted genetic modifications for improved cryopreservation outcome. Importantly, beyond expanding access to the premier biomedical research model, *Drosophila melanogaster*, we believe our work has compelling long-term implications for cryopreservation of still other important invertebrate species, providing important tools for mitigating decline in insect species abundance that can have far ranging impact on the food web[30]. More immediately, our success with *Drosophila* will inform efforts to develop successful cryopreservation of *Anopheles* mosquitoes, significantly contributing to the advancement of malaria research.

## Methods

### Stock maintenance.
Flies were maintained in *Drosophila* bottles (6 oz) at room temperature (24.2 ± 0.5 °C, Supplementary Fig. 2). Adults were removed from the bottle after 5–7 days. Fly food was prepared with the same recipe used by the Bloomington Stock Center. Specifically, BSDC Cornmeal Food was prepared using water, yeast, soy flour, yellow cornmeal, agar, light corn syrup, and propionic acid following the instructions listed on the website of Bloomington Stock Center[31].

The designation and genotype of stocks used in this study are provided in Supplementary Table 4.

### Cryopreservation protocol

*Step 1. Embryo collection and staging.* On day 1, a total of 700–1200 flies at the age of 1–4 days old were used to collect embryos at room temperature. Usually, four bottles of flies were used, eight or more bottles were used if needed. Flies were placed in an empty *Drosophila* bottle with a mesh cloth as a cap (Fig. 1a). Embryos were collected in 1 h period on a grape juice plate (diameter 63 mm) smeared with yeast paste. The first-hour collection served as an egg cleanup procedure for the female flies and these eggs were abandoned. Disturbance of flies was minimized during embryo collection. Grape juice plates with collected embryos were labeled with the end time point of collection, for instance, 3 pm was used to label the collection from 2 pm to 3 pm. Embryos were placed in a temperature incubator at 20.1 ± 0.05 °C (Heratherm, Thermo Scientific) until reaching the desired stage for cryopreservation. 20 °C was selected so that optimal embryo age for cryopreservation will be achieved during a normal work hour on the following day. In this work, embryo collection occurred in the afternoon and usually 2–4 collections were performed. In total, 200–600 embryos were collected and cryopreserved for each experiment depending on the *Drosophila* strain.

To stage the embryos on day 2, for instance, the embryo collection labeled as 3 pm on day 1 would reach 22 h old at 1 pm on day 2.

*Step 2. Dechorionation and permeabilization.* On day 2, embryos were washed off from the grape juice plate into a nylon mesh basket and dechorionated in 50% bleach for 2–4 min (Supplementary Fig. 1). After rinsing with running tap water for 1–2 min to remove excess bleach, embryos along with the mesh basket were briefly blotted on paper towel and placed in the cryobuffer (20 mM NaCl, 2.7 mM KCl, 10 mM $Na_2HPO_4$, 1.8 mM $KH_2PO_4$, 4 mM $MgCl_2$, 13 mM $MgSO_4$, 60 mM Glycine, 60 mM Glutamic acid, and 5 mM Malic acid, pH 6.8, sterilized by filtration) in a 35 mm petri dish. Embryos were examined under a dissecting microscope to confirm the removal of chorions. In addition, the gut morphology was evaluated to verify the embryo stage (Fig. 1b).

Before permeabilization, ~4 ml isopropanol, a mixture of D-limonene and heptane (1: 4 v/v), and heptane alone were added to three separate 35 mm glass petri dishes in a fume hood. A mesh basket was used to transfer the embryos from one solution to another. Specifically, the mesh basket was lifted from the cryobuffer and blotted on a paper towel to remove as much liquid as possible, followed by a 5–10 s dip in isopropanol until all embryos sank to the bottom. Then, the mesh basket with embryos inside was blotted on a paper towel several times to remove excess isopropanol. The embryos and mesh basket were then dried by blowing humid air (i.e, using mouth) until the mesh became see-through (Supplementary Fig. 29). This step is designed to remove the water on the embryo thereby allowing subsequent exposure to the organic solvent. It is critical to remove the isopropanol by drying since we noticed that the combination of isopropanol with heptane was toxic to the embryos. Next, the mesh basket was placed in the D-limonene and heptane mixture for 10 s to permeabilize the embryo. Similarly, after blotting on a paper towel, the mesh basket was placed in heptane for 5 s to remove the D-limonene around the embryo as D-limonene cannot be easily removed by evaporation. Finally, heptane was removed by air drying and the permeabilized embryos along with the mesh basket were placed back into the cryobuffer. The whole permeabilization process usually takes 1–2 min.

*Step 3. CPA loading and dehydration.* Right after permeabilization, a brush was used to break up clumps into individual embryos floating as a monolayer with minimal overlap (Fig. 1b). The mesh basket was blotted and then placed in 13 wt% ethylene glycol (EG) solution prepared with cryobuffer in a 35 mm petri dish. The embryos should remain floating in order to maintain access to oxygen. After 3 min, "wrinkles" on the embryo surface were observed under a dissecting microscope, indicating volumetric shrinkage (i.e., losing water) in response to higher external osmolarity (Fig. 1c). The percentage of embryos that shrunk was recorded. The 13 wt% EG petri dish was then placed in a humid chamber. At 25 min, embryos were inspected under a dissecting microscope to confirm that they swelled back to their original shape, indicating EG had entered the embryos. The percentage of embryos that swelled back was recorded. Usually, if the embryos were at the correct stage and properly permeabilized, >90% of the embryo would shrink and swell in 13 wt% EG (Fig. 1c).

Next, the mesh basket was blotted and placed in 39 wt% EG + 9 wt% sorbitol solution prepared in cryobuffer on ice (i.e., ~4 °C) for 9 min. This step dehydrates the embryos (i.e., water loss) thereby elevating the intra-embryonic EG concentration to favor vitrification during cooling and avoidance of devitrification during the rewarming processes. In general, 5–6 ml dehydration CPA was used in a 35 mm petri dish.

*Step 4. Transfer to the cryomesh*. After 9 min dehydration, a dry cryomesh was used to press the floating dehydrated embryos into the CPA solution from the top (Supplementary Fig. 1). Nearly all of the embryos stayed attached to the cryomesh after lifting the cryomesh out of the CPA solution. A paper towel was used to wick the majority of the remaining CPA solution on the cryomesh from the opposite side without the embryos. The wicking process should be done within 20 s as elevated temperature may increase CPA toxicity therefore leading to lower survival.

Assuming a medium packed monolayer of embryos (i.e, embryos occupy 30% of the total mesh area) and each embryo occupies 0.07 mm$^2$ (=3.14 * embryo half length * embryo half width = 3.14 * 0.25 mm* 0.09 mm), a 2 cm * 2 cm size mesh can accommodate 1714 (=20 mm * 20 mm * 0.3/0.07) embryos.

*Step 5. Vitrification and rewarming*. The cryomesh with dehydrated embryos was quickly plunged into liquid nitrogen. At this stage the embryos are cryopreserved and can be stored in liquid nitrogen until future use. To rewarm the embryos, the cryomesh was rapidly submerged into 30 wt% sucrose solution prepared in the cryobuffer (~40 ml solution in a 50 ml beaker) at room temperature while avoiding agitation. The 30 wt% sucrose was chosen to maintain the flattened embryo shape to avoid rapid rehydration and detachment of the embryos from the cryomesh.

*Step 6. CPA unloading and embryo culture*. After a few seconds (i.e., 5 s) in 30 wt% sucrose, the cryomesh along with the embryos were transferred to 15 wt% sucrose prepared in the cryobuffer for 3 min, followed by transfer to cryobuffer for 20 min to finally remove all of the intra-embryonic CPA. Finally, the embryos were transferred to a 35 mm petri dish filled with 1 ml Schneider medium using a brush. The petri dish was capped and placed in a humid chamber overnight.

*Step 7. Larvae hatch and adult eclosion*. On day 3, hatched larvae were transferred in the morning from the medium to food vials (15 × 95 mm shell vial). Embryo hatch rate was calculated using the ratio of hatched larvae to total embryos. The food vials with larvae were kept at room temperature. After 15 days, larvae to adult rate was calculated using the ratio of emerged adults to total larvae in the vials.

**Cooling and warming rate measurement**. To measure the cooling and warming rates of the cryomesh method, a bare wire type T thermocouple (unsheathed fine gauge thermocouples, wire diameter is 50 mm, OMEGA) and an oscilloscope were used. To test different cryogens, slush nitrogen was prepared by pulling vacuum to cool the liquid nitrogen until slush was formed. The thermocouple was glued to the cryomesh and the temperature was recorded during cooling and warming of the mesh alone. In addition, dehydrated embryos were collected and placed in contact with the thermocouple on the mesh to obtain the corresponding cooling/warming rates for a loaded mesh with CPA solution removed (Fig. 3). We also measured the cooling/warming rates with CPA solution remaining on the cryomesh (Fig. 3). Cooling and warming rates were calculated to represent rates during cooling and warming in the temperature zone from −140 °C to −20 °C using Microsoft Excel 2016. Importantly, the CPA solutions and CPA loaded *Drosophila* embryos will be in a glassy phase at −140 °C.

**Warming rate modeling**. We used COMSOL 5.4 to simulate the warming rate of embryos using the cryomesh method. Two extreme conditions were considered: (1) minimal contact between dehydrated embryo and the cryomesh, and (2) maximal contact between the dehydrated embryo and the cryomesh (Fig. 3). The cross-section of nylon fibers was set as 150 × 80 µm, aperture was 200 µm, the length and width of the embryo were 500 µm and 180 µm respectively based on direct measurements. To estimate the thickness of dehydrated embryos, we first measured the weight of 532 dehydrated embryos to be 2.6 mg (Fig. 3). The weight of a single dehydrated embryo was then calculated to be 4.9 µg. Assuming the dehydrated embryo density to be the density of embryo solid content (1.37 g/ml)[32,33], we estimate the thickness of dehydrated embryo to be 50 µm. As the thermal properties of dehydrated embryos are unknown, we used temperature-dependent thermal properties of CPA based on previous publications[27,34]. For the nylon mesh, the density was set to be 1.15 g/ml, temperature-dependent thermal conductivity and heat capacity obtained from National Institute of Standards and Technology (NIST)[35–37]. Convective heat flux was used as the boundary conditions with convective heat transfer coefficient set as 300 W/(m$^2$*K)[38,39]. Warming rates at different cross-sections through the center point of embryos were compared for two extreme conditions.

In addition, we modeled the warming rate of the methods used in previous publications. Specifically, polycarbonate filter with 10 µm pore size (item # F10013–MB, SPI Supplies) and copper grid for electron microscope with 200 µm aperture (item # G100-Cu, Electron Microscopy Sciences) (Table S1). We assumed the CPA solution around the embryos to be 250 µm thick[8,9,14,15,40].

**PCR**. To track the single-nucleotide polymorphism (SNP) of the strain M2 post cryopreservation, DNA was prepared from a single male adult. The fly was placed in an Eppendorf tube and crushed with a pipette tip in 20 µl solution (10 mM Tris-Cl, pH 8.2, 1 mM EDTA, 25 mM NaCl, and 200 µg/ml proteinase K). The tube was incubated at 65 °C for 20 min, then 95 °C for 5 min. PCR reaction was carried out using Go-Taq flexi DNA polymerase in following cycles: 1x cycle of 95 °C for 2

min, 35x cycles of 50 °C 2 min, 72 °C 3 min and 94 °C 1 min, 1 cycle of 50 °C 2 min and 72 °C 10 min. Primers used were primer 1: 5′-ACG ATC TGG ATC CAG TCG-3′ and primer 2: 5′-GGA ATT CTT GTC TTC CTT GAC GCT CG-3′.

**Electron microscopy (EM)**. Embryos before and after permeabilization treatment were used for the EM imaging. Similar sample preparation procedures were used as previously reported[41,42]. Specifically, embryos were first ruptured with small holes and fixed in 2% paraformaldehyde and 2% glutaraldehyde in 0.1 M sodium cacodylate buffer (pH 7.4) at 4 °C overnight. Embryos were then washed with 0.1 M sodium cacodylate buffer three times (each time 15 min), 1 h in 4% osmium tetroxide, followed by rinse in distilled water. Next, embryos were dehydrated in a series of ethanol solution and three changes of propylene oxide. The resin was added to infiltrate and polymerize inside the embryos in a 60 °C oven overnight. Finally, ultrathin sections were cut. Uranyl acetate and lead citrate were used to stain the sections for EM imaging. ImageJ 1.52a was used to label the image (Supplementary Fig. 5).

**Statistics**. For plots with two dependent variables, for instance, hatch rate and adult rate, or cooling rate and warming rate, multivariate analysis of variance (MANOVA) and Tukey's post hoc were used for statistical analysis in software SPSS Statistics 23.

For Fig. 3b, paired two-tailed student's t-test was used in Origin 2018. "ns" represents the difference is not statistically significant ($p > 0.05$), $*p \leq 0.05$, $**p \leq 0.01$, $***p \leq 0.001$, $****p \leq 0.0001$. The $p$ value of all the statistical analysis is provided in an Excel file as the Supplementary materials.

**Reporting summary**. Further information on experimental design is available in the Nature Research Reporting Summary linked to this paper.

## Data availability

All relevant data supporting the key findings of this study are available within the article and its Supplementary Information files or from the corresponding authors upon reasonable request. The *Drosophila* strains that we have used in this report are also available from the corresponding authors upon reasonable request. A reporting summary for this article is available as a Supplementary Information file. Source data are provided with this paper.

## Code availability

The COMSOL files used in the "warming rate" modeling is available from the corresponding authors upon reasonable request.

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

## Acknowledgements

We thank Dr. Amanda Neisch for productive discussions surrounding the genetics and variation in cryopreservation. We thank Zonghu Han and Anthony Chen for serving as volunteers to test the robustness of the developed cryopreservation protocol. We thank Dr. Michael Etheridge for his critical reading and comments to the manuscript. Funding from the US National Institutes of Health (NIH) (1R21OD028758) is gratefully acknowledged. T.H. further acknowledges NIH GM R01GM044757 and J.B. acknowledges NSF EEC 1941543. L.Z. acknowledges the support of a Doctoral Dissertation Fellowship at the University of Minnesota.

## Author contributions

L.Z., M.L., T.H., and J.B. conceived and designed the study. L.Z. and M.L. performed data collection. L.Z., M.L., T.H., and J.B. analyzed and interpreted data. L.Z., M.L., T.H., and J.B. wrote the manuscript.

## Competing interests

L.Z., M.L., T.H. and J.B. have provisional patent applications (Serial No. 63/136,366) relevant to this study. The authors declare competing interests.

## Additional information

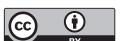

