## [Peer Review File · Nature Communications]

Reviewers' Comments:

Reviewer #1:

Remarks to the Author:

The lack of robust methodology to preserve *Drosophila Melangaster* has been a pervasive hindrance for geneticists that rely heavily on this model organism. The absence of an effective methodology to preserve mutants invariably leads to loss of valuable assets, and this challenge compounds with technical and cost burdens.

This manuscript presents a very straightforward and robust methodology that builds upon decades of knowledge in cryobiology, while also introducing simple in concept, yet highly effective innovations that improve outcomes and will facilitate adoption of this technique by other users.

In this expertly written and meticulously organized paper, the authors present a very thoughtful and systematic tour de force through the many variables that could influence preservation outcome. This includes staging of eggs, culture techniques pre- and post-preservation, as well as the many variables specific to cryopreservation processing, including permeabilization strategy, choice and concentration of CPA, loading time and temperature, and cooling and warming rates. All experiments were presented clearly and effectively with the appropriate statistics, and all conclusions were well justified. The figures, tables, and illustrations are also of very high quality, and organized in a way that supports ease of understanding, and reproduction of methodologies. Overall this is a clear, comprehensive, and highly impactful paper that meets a long overdue need in the field.

The methods described in this paper are very likely to be immediately adopted by researchers that use the *Drosophila Melangaster* model, transforming the range of research that will be possible. The innovations presented will also have widespread impact on the preservation of embryos of other highly relevant recalcitrant organisms, such as disease-bearing mosquito strains.

Small suggestions:

Abstract: 'novel innovations' is redundant. Better to use one word or the other.

Fig 1, the degree symbol is missing in legend.

General: What is the basis for the selection of components in the 'cryo-buffer'? Is this a common solution used in the field or was it optimized/conceived for use in this study?

Reviewer #2:

Remarks to the Author:

This paper reports the successful implementation of embryo freezing in *Drosophila*.

Cryopreservation is a well established method for many organisms, however, the approach has lagged behind for *Drosophila*. Having the ability to freeze fly embryos has huge implications considering the number of fly lines available for this key model organism. Previous studies have provided some evidence that freezing was possible but the method never took off because of difficulties in implementations and variable success rates. In this impressive paper the authors show that they can get 'good enough' survival rates without any special equipment and they do a thorough job exploring the different variables. This paper will revive interest from the community in fly cryopreservation and should be published as soon as possible.

Minor comments:

In Figure 2, it should be made clear that the red/blue hatch rate/adult rate labeling applies to the whole figure.

It should be indicated in the supplemental file that in addition to Figure S3 red boxes = embryo hatch rate and blue boxes = adult rate in S4, S6, S7, S8, S9, S14, S15, S16, S17, S18, S19, S20, S21, S22, S23, S24, S25, S27.

On a more general note about cryopreservation, one doesn't know which strains will have low survival rates, so for each one a lab needs to do the following:

- 1) test survival in short term cryopreservation
- 2) outcross if necessary
- 3) repeat 1 and 2
- 4) do long term cryopreservation

Especially for mutant alleles maintained over balancers and balancer stocks, which have the lowest survival rate (see Figure 4) this seems like a lot of up-front work to determine optimal conditions, as each stock must be frozen and thawed first to find out if the protocol works, then troubleshooting, then frozen again. The authors should discuss this more extensively.

Reviewer #3:

Remarks to the Author:

In this manuscript Zhan et al. established a protocol that allows long-term cryopreservation of wildtype and mutant *Drosophila* stocks. They optimized embryo collection parameters, embryo treatments to load and unload Cryo Protective Agents (CPA), CPA contents, and reviving the cryopreserved embryos. They show that their optimized protocol does not require extensive special equipment and can be implemented on independent strains by relatively untrained operators.

The lack of a simple long-term cryopreservation protocol has been a handicap for *Drosophila* biology. Hence, a lot of effort and resources need to be dedicated to stock keeping in individual laboratories and in stock centers. Hence, cryopreservation research has been the focus of numerous *Drosophila* white papers. The methods established in this manuscript are relatively easy to implement and the authors nicely demonstrate the robustness and limits of the protocol with emphasis on where there may be need for troubleshooting. The authors show that the methods are sensitive and there is variation in recovery rate among different *Drosophila* strains. However the embryo to adult rate after cryopreservation is around 20% in all but one of the stocks, which is very promising for widescale applicability of the method. Also, the manuscript is written in a detailed manner that will facilitate the implementation by any lab. Hence, this manuscript will be extremely important to the whole *Drosophila* research community. I only have one minor issue.

Minor issue:

-The genotypes of the stocks are not accurate, not in italics, and should be listed in materials and methods and in supplementary information. The precise genotype of the used strains should be provided.

In summary, the manuscript is well written and describes a very important method that will help the whole *Drosophila* field.

February 24th, 2021

Dear Reviewers,

We deeply appreciate the time and thoughtful consideration that the Reviewers have given to our manuscript towards increased clarity and impact. We have provided below a point-by-point response to all the comments (*in italics*) from the Reviewers. We note in **blue text** any changes made to the manuscript and Supplemental Information in response to the review.

Response to Reviewer #1

The lack of robust methodology to preserve Drosophila Melanogaster has been a pervasive hindrance for geneticists that rely heavily on this model organism. The absence of an effective methodology to preserve mutants invariably leads to loss of valuable assets, and this challenge compounds with technical and cost burdens.

This manuscript presents a very straightforward and robust methodology that builds upon decades of knowledge in cryobiology, while also introducing simple in concept, yet highly effective innovations that improve outcomes and will facilitate adoption of this technique by other users.

In this expertly written and meticulously organized paper, the authors present a very thoughtful and systematic tour de force through the many variables that could influence preservation outcome. This includes staging of eggs, culture techniques pre- and post-preservation, as well as the many variables specific to cryopreservation processing, including permeabilization strategy, choice and concentration of CPA, loading time and temperature, and cooling and warming rates. All experiments were presented clearly and effectively with the appropriate statistics, and all conclusions were well justified. The figures, tables, and illustrations are also of very high quality, and organized in a way that supports ease of understanding, and reproduction of methodologies. Overall this is a clear, comprehensive, and highly impactful paper that meets a long overdue need in the field.

The methods described in this paper are very likely to be immediately adopted by researchers that use the Drosophila Melanogaster model, transforming the range of research that will be possible. The innovations presented will also have widespread impact on the preservation of embryos of other highly relevant recalcitrant organisms, such as disease-bearing mosquito strains.

Response: We thank the reviewer for the enthusiastic comments and encouragement.

Small suggestions:

1. *Abstract: ‘novel innovations’ is redundant. Better to use one word or the other.*

Response: We thank the reviewer for pointing this out. We removed “novel” in the sentence.

Page 1 of the manuscript:

We present innovations for embryo permeabilization, cryoprotectant agent loading, and rewarming.

2. Fig 1, the degree symbol is missing in legend.

Response: We thank the reviewer for the careful read. However, we believe that we included all three degree symbols for temperature (highlighted in yellow) in the legend of Fig. 1 as shown below.

Page 11-12 of the manuscript:

Fig. 1. Schematic overview of cryopreservation procedures for *Drosophila melanogaster* embryos and detailed pictorial illustration for critical steps. (A) On day 1, embryos were collected on a grape juice plate for 1 hour period at room temperature (24 °C), then placed in a 20 °C incubator until reaching desired stage for cryopreservation. On day 2, embryos were first dechorionated and permeabilized, followed by CPA loading and dehydration. The cryomesh was used to pick up the dehydrated embryos for vitrification, storage, rewarming and CPA unloading. Afterwards, embryos were cultured in Schneider's medium overnight. On day 3, hatched larvae were transferred to food vials until adult emergence. **(B)** Images of embryo gut morphology under dissecting and compound microscopes after different incubation time at 20 °C. **(C)** Images of embryos at different steps during cryopreservation. From left to right, impermeable and permeable embryos after 5 min in 0.1 % rhodamine B; permeable embryos first shrunk then swelled in 13 wt% EG; flattened embryos after dehydration; vitrified embryos appeared transparent in LN₂ (two of them were outlined in red), inset is a crystallized embryo (i.e., failure). Scale bars in compound microscope images are 100 μm, in dissecting microscope images are 500 μm.

3. *General: What is the basis for the selection of components in the 'cryo-buffer'? Is this a common solution used in the field or was it optimized/conceived for use in this study?*

Response: The 'cryo-buffer' was conceived for use in this study by modifying the recipe of Schneider's *Drosophila* medium to 1) maintain isotonic osmolarity and 2) reduce risk of contamination.

Response to Reviewer #2

*This paper reports the successful implementation of embryo freezing in *Drosophila*. Cryopreservation is a well established method for many organisms, however, the approach has lagged behind for *Drosophila*. Having the ability to freeze fly embryos has huge implications considering the number of fly lines available for this key model organism. Previous studies have provided some evidence that freezing was possible but the method never took off because of difficulties in implementations and variable success rates. In this impressive paper the authors show that they can get 'good enough' survival rates without any special equipment and they do a thorough job exploring the different variables. This paper will revive interest from the community in fly cryopreservation and should be published as soon as possible.*

Response: We thank the reviewer for the positive comments.

Minor comments:

1. *In Figure 2, it should be made clear that the red/blue hatch rate/adult rate labeling applies to the whole figure.*

Response: We thank the reviewer for this important suggestion. We added a clear statement in the legend of Figure 2 to call this out.

Page 13 of the manuscript:

The red/blue hatch rate/adult rate labeling applies to the entire Figure 2.

2. *It should be indicated in the supplemental file that in addition to Figure S3 red boxes = embryo hatch rate and blue boxes = adult rate in S4, S6, S7, S8, S9, S14, S15, S16, S17, S18, S19, S20, S21, S22, S23,S24, S25, S27.*

Response: We thank the reviewer for this comment. We modified the legend of Figure S3 to reflect this.

Page 4 of the supplemental file:

In (c), red boxes present embryo hatch rate (i.e., embryo to larvae) and blue boxes represent adult rate (i.e., resulting larvae to adults). The same labeling applies to Figs. S4, S6, S7, S8, S9, S14, S15, S16, S17, S18, S19, S20, S21, S22, S23,S24, S25, S27.

3. *On a more general note about cryopreservation, one doesn't know which strains will have low survival rates, so for each one a lab needs to do the following:*

1) test survival in short term cryopreservation

2) outcross if necessary

3) repeat 1 and 2

4) do long term cryopreservation

Especially for mutant alleles maintained over balancers and balancer stocks, which have the lowest survival rate (see Figure 4) this seems like a lot of up-front work to determine optimal conditions, as each stock must be frozen and thawed first to find out if the protocol works, then troubleshooting, then frozen again. The authors should discuss this more extensively.

Response: We thank the reviewer for this valuable suggestion. In this response, we define the following terms:

Intrinsic embryo to adult survival = Percentage of adults developed from untreated embryos,

Post thaw embryo to adult survival = Percentage of adults developed from cryopreserved embryos (normalized to the intrinsic embryo to adult survival).

We added a more extensive discussion regarding the adoption of our protocol by the *Drosophila* community as shown below and in Page 7 of the manuscript:

a. *The optimized protocol applies directly to the majority of tested strains. The post thaw embryo to adult survival is > 9% for 96% of the tested strains, and >15% for 64% of the tested strains (Table S4, Figure 4a). To cryopreserve the mutant strains with low intrinsic embryo to adult survival (i.e., 13% for strain S9), one could start with more embryos in order to obtain sufficient adults post thaw. In this case, one only needs to perform 1) and 4) as listed by the reviewer.*

- b. *Outcrossing is a tool to improve post thaw survival.* For mutant strains with both low intrinsic and post thaw embryo to adult survival, we showed that the survival can be improved by outcrossing to stocks with high post thaw survival (i.e., strain S1 to NS1). Indeed, these steps (i.e., 2 & 3 listed by the reviewer) may increase the up-front work. However, if successful, we believe the benefits will outweigh the up-front efforts.
- c. *Further study to investigate the genetic factors.* Further investigation focusing on the genetic aspect of mutant stocks could lead to more insights on the strain-dependent post thaw survivals. This could help to potentially identify the gene(s) that contributes to the variation in post thaw survival. If so, the up-front work (i.e., 2 & 3 listed by the reviewer) can be greatly reduced.

Page 7 of the manuscript:

Further investigation focusing on the genetic aspect could potentially identify the gene(s) that contributes to the variation in post thaw survival, therefore allowing targeted genetic modifications for improved cryopreservation outcome.

Response to Reviewer #3

In this manuscript Zhan et al. established a protocol that allows long-term cryopreservation of wildtype and mutant Drosophila stocks. They optimized embryo collection parameters, embryo treatments to load and unload Cryo Protective Agents (CPA), CPA contents, and reviving the cryopreserved embryos. They show that their optimized protocol does not require extensive special equipment and can be implemented on independent strains by relatively untrained operators.

The lack of an simple long-term cryopreservation protocol has been a handicap for Drosophila biology. Hence, a lot of effort and resources need to be dedicated to stock keeping in individual laboratories and in stock centers. Hence, cryopreservation research has been the focus of numerous Drosophila white papers. The methods established in this manuscript are relatively easy to implement and the authors nicely demonstrate the robustness and limits of the protocol with emphasis on where there may be need for troubleshooting. The authors show that the methods are sensitive and there is variation in recovery rate among different Drosophila strains. However the embryo to adult rate after cryopreservation is around 20% in all but one of the stocks, which is very promising for widescale applicability of the method. Also, the manuscript is written in a detailed manner that will facilitate the implementation by any lab. Hence, this manuscript will be extremely important to the whole Drosophila research community. I only have one minor issue.

Response: We thank the reviewer for the nice summary of our work and positive feedbacks.

Minor issue:

- 1. The genotypes of the stocks are not accurate, not in italics, and should be listed in materials and methods and in supplementary information. The precise genotype of the used strains should be provided.*

Response: We have updated genotypes of the stocks, with italics, in materials and methods and in supplementary information. The genotype in Figure 4a is now removed.

Page 18 of the manuscript:

Below are the designation and genotype (i.e., in parenthesis) of stocks used in this study:

Designation	Genotype information
OR	(Oregon-R)
WC	(w [1118])
WC1	(w [1118] derivative; outcrossed to isogenize a single X chromosome from w [1118])
WC1b	(w [1118] derivative; 2 nd independent stock from outcross of w [1118] to isogenize for single X chromosome)
WC1.1	(w [1118] derivative; new stock from outcross of WC1, only X chromosome of WC1 is maintained)
WC2	(w [1118] derivative; outcrossed to isogenize a single 2 nd chromosome)
WC3	(w [1118] derivative; outcrossed to isogenize a single 3 rd chromosome)
WC3b	(w [1118] derivative; 2 nd independent stock from outcross of w [1118] to isogenize for single 3 rd chromosome)
GFP	(y [1] w [*]; PBac { y [+ mDint2] w [+ mC]= Dfd-EGFP.S } VK00037)
M2	(w [1118] derivative with a T to G SNP at the position of bp568 in coding sequences of CG1938)
M2-3b	(w [1118] derivative; outcrossed to isogenize a single 3 rd chromosome from M2)
S1	(w ; Sp / CyO)
S2	(po ros/w FM6)
S3	(Dhc64C ⁶⁻¹² , P { neoFRT } 80B/TM3)
S4	(w [1118]; Sp / CyO ; TM2/TM6)
S5	(elav-GAL4 ANF-GFP ; TM3/TM6)
S6	(Sp-EM6/FM7-GFP)
S7	(DhcGFP11-3/TM3 Sb)
S8	(TM3 Sb/TM6B Tb)
S9	(w ; Bl [1]/ CyO ; TM2/TM6 UAS-GAL80)
S10	(w ; Bl [1]/ CyO ; TM2/TM6)
S11	(bActβ80/UAC-D-GFP)
S12	(po ros/w FM6 ; Sp / CyO)
NS1	(y [1] w [*]; Sp / CyO)
yw1	(y [1] w [*])

Page 35 of the Supplementary materials:

** Details of the genotype can be found in the Methods under the section of Stock maintenance.

2. *In summary, the manuscript is well written and describes a very important method that will help the whole Drosophila field.*

Response: We thank the reviewer for the encouragement.